# Role of Copeptin and hs-cTnT to Discriminate AHF from Uncomplicated NSTE-ACS with Baseline Elevated hs-cTnT—A Derivation and External Validation Study

**DOI:** 10.3390/cells12071062

**Published:** 2023-03-31

**Authors:** Stephan von Haehling, Matthias Müller-Hennessen, Tania Garfias-Veitl, Alina Goßling, Johannes T. Neumann, Nils A. Sörensen, Paul M. Haller, Tau Hartikainen, Jörn Ole Vollert, Martin Möckel, Stefan Blankenberg, Dirk Westermann, Evangelos Giannitsis

**Affiliations:** 1Department of Cardiology and Pneumology, University of Göttingen Medical Center, 37075 Göttingen, Germany; stephan.von.haehling@med.uni-goettingen.de; 2German Center for Cardiovascular Research (DZHK), Partner Site Göttingen, 37075 Göttingen, Germany; 3Department of Cardiology, Angiology and Pneumology, Heidelberg University Hospital, 69120 Heidelberg, Germany; matthias.mueller-hennessen@med.uni-heidelberg.de (M.M.-H.); evangelos.giannitsis@med.uni-heidelberg.de (E.G.); 4Department of Cardiology, University Heart & Vascular Center Hamburg, Martinistrasse 52, 20246 Hamburg, Germany; a.gossling@uke.de (A.G.); jo.neumann@uke.de (J.T.N.); n.soerensen@uke.de (N.A.S.); p.haller@uke.de (P.M.H.); s.blankenberg@uke.de (S.B.); 5German Center of Cardiovascular Research (DZHK), Partner Site Hamburg/Kiel/Lübeck, 10115 Hamburg, Germany; 6University Heart Center Freiburg Bad Krozingen, Department of Cardiology and Angiology, University Freiburg, 79110 Freiburg, Germanydirk.westermann@uniklinik-freiburg.de (D.W.); 7BRAHMS GmbH Deutschland, 10785 Berlin, Germany; joern.vollert@thermofisher.com; 8Department of Emergency Medicine Campus Charité Mitte, Virchow-Klinikum and Department of Cardiology, Charité-Universitätsmedizin Berlin, 13353 Berlin, Germany; martin.moeckel@charite.de

**Keywords:** copeptin, high-sensitivity cardiac troponin, outcomes, acute heart failure, acute coronary syndrome

## Abstract

Background: In light of overlapping symptoms, discrimination between non-ST-elevation (NSTE) acute coronary syndrome (ACS) and acute heart failure (HF) is challenging, particularly in patients with equivocal clinical presentation for suspected ACS. We sought to evaluate the diagnostic and prognostic properties of copeptin in this scenario. Methods: Data from 1088 patients from a single-center observational registry were used to test the ability of serial high sensitivity cardiac troponin T (hs-cTnT)—compared to copeptin, or a combination of copeptin with hs-cTnT—to discriminate acute HF from uncomplicated non-ST-elevation myocardial infarction (NSTEMI) and to evaluate all-cause mortality after 365 days. Patients with STEMI, those with unstable angina and either normal or undetectable hs-cTnT concentrations were excluded. The findings were validated in an independent external NSTE-ACS cohort. Results: A total of 219 patients were included in the analysis. The final diagnosis was acute HF in 56 and NSTE-ACS in 163, with NSTEMI in 78 and unstable angina having stable elevation of hs-cTnT >ULN in 85. The rate of all-cause death at 1 year was 9.6% and occurred significantly more often in acute HF than in NSTE-ACS (15 vs. 6%, *p* < 0.001). In the test cohort, the area under the receiver operator curve (AUC) for the discrimination of acute HF vs. NSTE-ACS without HF was 0.725 (95% confidence interval [CI] 0.625–0.798) for copeptin and significantly higher than for hs-cTnT at 0 h (AUC = 0.460, 0.370–0.550) or at 3 h (AUC = 0.441, 0.343–0.538). Copeptin and hs-cTnT used either as continuous values or at cutoffs optimized to yield 90% specificity for acute HF were associated with significantly higher age- and sex-adjusted risk for all-cause mortality at 365 days. The findings from the test cohort were consistently replicated in the independent external NSTE-ACS validation cohort. Conclusions: High concentrations of copeptin in patients with suspected NSTE-ACS and equivocal clinical presentation suggest the presence of acute HF compared to uncomplicated NSTE-ACS and are associated with higher rates of all-cause death at 365 days.

## 1. Introduction

The clinical presentation of patients with acute coronary syndromes (ACS) is heterogenous, and many cases present with atypical symptoms, dyspnea, unspecific ECG abnormalities or normal ECG, and elevated cardiac troponin values. The discrimination of non-ST-elevation (NSTE) ACS from other life-threatening differential diagnoses including heart failure (HF) or pulmonary embolism is challenging when the clinical and laboratory criteria of the fourth universal definition of myocardial infarction are applied [1]. Since only about 10% of patients presenting with chest pain to emergency departments (ED) are finally diagnosed with ACS, early markers to narrow the differential diagnosis down are urgently needed. Neither symptoms nor ECG criteria can help in this regard [2,3].

While biomarkers have been found useful to diagnose acute or chronic heart failure in patients presenting with suspected heart failure, there is sparse evidence on the role of additional biomarkers beyond the routine measurement of cardiac troponins among dyspneic patients presenting with suspected ACS, a setting where natriuretic peptides are not recommended routinely.

Due to its tissue specificity, cardiac troponin is regarded as the preferred biomarker for the diagnosis of acute myocardial infarction (MI), but it lacks clinical specificity to allow a discrimination between the various cardiovascular reasons for acute or chronic myocardial injury including acute heart failure, myocarditis or pulmonary embolism [4]. Cardiac imaging including echocardiography and computed tomography (CT) are recommended to facilitate the diagnosis and differential diagnosis [5]. However, both methods are underutilized in the ED [6].

While natriuretic peptides are recommended for the diagnosis of suspected acute or chronic HF [7], they provide some prognostic but no diagnostic information in suspected ACS [8,9]. According to guidelines on NSTE-ACS published by the European Society of Cardiology (ESC), only copeptin may be considered for the instant rule-out of patients with suspected NSTE-ACS at low-to-intermediate risk when hs-cTn assessment is not available [1,4], because “the added diagnostic value of copeptin to conventional (less sensitive) cardiac troponin assays is substantial” [8]. In addition, copeptin has been reported to add prognostic information in ACS [10,11,12] and HF [13], independent of that contained in the hs-cTn concentration or the GRACE score.

Given that hs-cTn values are almost always elevated in patients with acute HF or with NSTEMI, surprisingly little information is available as to whether copeptin may help to discriminate acute HF from uncomplicated NSTE-ACS. The measurement of copeptin together with hs-cTn has been found particularly useful to rule out NSTEMI [12,14,15] and to identify patients at high risk of death across a multitude of acute settings, including acute HF [13]. Therefore, the present analysis in patients with a primary working diagnosis of suspected ACS but not suspected acute heart failure sought to investigate the ability of copeptin added to hs-cTnT to improve the diagnostic discrimination between acute HF and uncomplicated NSTE-ACs and to predict 30-day mortality.

## 2. Methods

### 2.1. Derivation Cohort

Patients with suspected ACS were recruited between August 2014 and July 2017 based on a broad range of symptoms suggestive of ACS as part of an observational registry (MIR) at the Chest Pain Unit of the University Hospital Heidelberg, Germany. Along with troponin, suspicion of ACS was based on a combination of clinical symptoms, presence of atherosclerotic risk factors, ECG, previous history of CAD/previous MI/previous PCI or CABG, dyspnea and atypical chest pain. Patients with unequivocal signs or symptoms of acute heart failure at low pre-test probability for NSTE-ACS were not enrolled. The study cohort has been described in detail earlier [16]. In brief, NSTE-ACS including NSTEMI (type 1 MI or type 2 MI) and unstable angina with stable elevated cardiac troponin was diagnosed according to the criteria of the third universal definition of myocardial infarction [17]. Patients with new or worsening symptoms without an elevation in hs-cTnT or without a relevant concentration change were diagnosed as having unstable angina. Type 2 MI was diagnosed in the presence of a rise and/or fall in cTn values with at least one value above the 99th percentile URL, and evidence of an imbalance between myocardial oxygen supply and demand unrelated to coronary thrombosis, requiring at least one clinical variable that suggests a context of myocardial ischemia. In contrast, type 1 MI represents the typical spontaneous MI characterized by plaque rupture/erosion with occlusive or non-occlusive thrombosis. Among these, patients with normal or undetectable hs-cTnT values were excluded from the analysis, as were patients with an ST-elevation MI (STEMI, *n* = 6). Acute HF with or without NSTE-ACS was diagnosed using a combination of clinical signs and symptoms with a value of N-terminal pro B-type natriuretic peptide (NT-proBNP) exceeding the age-adjusted rule-in cutoff [18]. Patients with NSTE-ACS complicated by acute HF based on any NT-proBNP value above the age-independent rule-out cutoff of 300 ng/L were excluded. In the NSTE-ACS cohort, we used NT-proBNP as the sex-independent rule-out cutoff. Patients with values of <300 pg/mL are unlikely to have an acute heart failure. On the contrary, we excluded patients with values > 300 pg/mL. The Global Registry of Acute Coronary Events (GRACE) score was calculated retrospectively in all patients using the GRACE 1.0 calculator [19]. The score was used as a continuous variable to characterize the overall risk of the NSTEMI or AHF cohort, and was not used to categorize individual patients into low, intermediate or high risk. The final diagnosis was retrospectively adjudicated by two independent cardiologists considering all clinical and diagnostic findings.

Follow-up was performed until at least 30 days after discharge via telephone calls, in written form, via the hospital information system, or by obtaining information on survival from the local residents’ registration offices. The study was carried out according to the principles of the Declaration of Helsinki. Each registry was approved by the local ethics committee of the University of Heidelberg (No. 3302003) and written informed consent was obtained from all patients.

### 2.2. Validation Cohort

Findings from the MIR cohort described above were validated using data from the prospective biomarkers in the Acute Cardiac Care (BACC) cohort study that enrolled consecutive patients with suspected acute MI who presented to the ED or chest pain unit of the University Hospital Hamburg-Eppendorf, Germany. Findings from the BACC study have been published earlier [20,21]. The final diagnosis was adjudicated in accordance with the fourth Universal Definition of MI and NSTEMI. Type 1 and type 2 were sub-classified according to the presumed pathomechanism. For the validation of our findings, we identified patients with NSTEMI type 1 and patients with acute HF. Patients with STEMI, unstable angina with stable normal or undetectable hs-cTnT values, and NSTEMI type 2 or type 4 were excluded.

### 2.3. Biomarker Analysis

Blood samples were collected in plasma tubes at presentation. Following centrifugation, samples were frozen at −80 °C for further analysis. Hs-cTnT was measured using the Elecsys^®^ Troponin T-high sensitive assay (Roche Diagnostics, Mannheim, Germany) on a Cobas e411 immunoassay analyzer. Limit of blank, limit of detection, 10% coefficient of variation (CV), and 99th percentile cut-off values were determined to be 3, 5, 13 and 14 ng/L [22,23]. Measurement of copeptin was performed with the copeptin proAVP assay on the KRYPTOR compact plus (BRAHMS GmbH, Hennigsdorf, Germany).

### 2.4. Statistical Analysis

Continuous variables are presented as median with interquartile range for a non-normal distribution or as means with 95% confidence intervals for normally distributed data. For the comparison of continuous parameters, the Mann–Whitney U-test was used, whereas a chi-square test was applied for categorical parameters. For prognostic assessment, all-cause mortality was assessed using Cox regression and plotted in Kaplan–Meier survival plots. Receiver operator characteristic (ROC) curves were used to determine the prognostic performance with area under the curves (AUCs) for the prediction of adverse outcomes. We used AUC to identify the cutoff that provided 90% or more specificity for the diagnostic discrimination between NSTE-ACS and acute HF. This cutoff was used as a dichotomous variable for its value to predict all-cause death. All hypothesis testing was two-tailed and *p*-values of <0.05 were considered statistically significant. Statistical analyses were performed using MedCalc 15.6 (MedCalc Software, Ostend, Belgium), Statistical Package for the Social Sciences (SPSS) version 26.0 (IBM, Armonk, NY, USA) and R: A language and environment for statistical computing version 4.0.3 (R Foundation for Statistical Computing, Vienna, Austria).

## 3. Results

The consort diagram (Figure 1) describes the composition of the final derivation cohort. Of 1088 patients screened, a total of 219 patients were eligible including 163 patients with uncomplicated NSTE-ACS and 56 patients with acute HF without NSTE-ACS. Rates of coronary angiography varied by ACS subtype and study center, between 85 and 95% for STEMI, 55 and 70% for NSTEMI and 20 and 40% for UAP. Baseline characteristics of the derivation cohort are listed in Table 1. Briefly, patients with acute HF differed significantly from patients with NSTE-ACS without HF for almost all clinical variables except for similar rates in cardiovascular risk factors, similar rates in previous MI, revascularization therapies, or stroke. Whilst the concentrations of copeptin were significantly higher in patients with acute HF (*p* < 0.001), concentrations of hs-cTnT at 0 h and at 3 h were similar. It is noteworthy that the maximum hs-cTnT concentrations were significantly higher in patients with NSTE-ACS than in those with acute HF (*p* = 0.022), and concentration changes between serial samples tended to be higher in patients with NSTE-ACS.

### 3.1. Discrimination between NSTE-ACS and Acute HF

In the entire cohort, median copeptin was 9.2 (IQR 4.9–17.7) pmol/L (mean: 20.6 ± 30.7 pmol/L). In the subgroup of patients with acute HF, the corresponding values were 17.1 (8.8–54.5) pmol/L (mean: 30.1 ± 29.2 pmol/L), with copeptin elevated above 10 pmol/L in 71.4% (Table 2). In the subgroup of patients with NSTE-ACS, the corresponding values were 7.7 (4–14.5) pmol/L (mean: 17.3 ± 30.6 pmol/L), with copeptin elevated above 10 pmol/L, which is the common rule-out cutoff for NSTE-ACS, in 36.8%.

In the entire cohort, median hs-cTnT at 0 h was 26 (17–47) ng/L (mean: 74.7 ± 194.2 ng/L). In the subgroup of patients with acute HF, the corresponding values were 25.5 (15–40.8) ng/L (mean: 46.5 ± 82.5), with hs-cTnT elevated above 14 ng/L in 80.4% (Table 2). In the subgroup of patients with NSTE-ACS, the corresponding values were 26 (18–47) ng/L (mean: 84.4 ± 219.3 ng/L). Hs-cTnT concentrations above the 99th percentile were present in 95.1%, with a median concentration of 27 ng/L (18–52) (Table 2). The ability of copeptin and hs-cTnT to discriminate acute HF from uncomplicated MI differed widely according to whether troponin or copeptin were used and by the time point of the blood draw (Figure 2). Whilst the area under the curve (AUC) for hs-cTnT at baseline (AUC: 0.460, 95% CI: 0.370–0.550, SE: 0.046) or at 3 h (AUC: 0.441, 95% CI: 0.343–0.538, SE: 0.050) was low, indicating poor performance, copeptin showed moderate discrimination with an AUC of 0.725 (95% CI: 0.652–0.798, SE: 0.037). The combination of copeptin and hs-cTnT increased the AUC, although the increase was not significant (delta AUC 0.08, *p* = 0.87).

### 3.2. Prediction of 365-Day Mortality Using hs-cTnT and Copeptin

In the derivation cohort, 21 deaths (9.6%) had occurred by 365 days. The mortality rate was higher among patients with acute HF than in those with NSTE-ACS (hazard ratio [HR] 8.4, 95% confidence interval [CI] 3.2–21.6, *p* < 0.001; 15 vs. 6 deaths; 26.79 vs. 3.68%). Compared to survivors after 365 days, patients who died had higher median baseline copeptin concentrations (31.6 (12.3–63.8) vs. 8.5 (4.4–16.1) pmol/L, *p* < 0.001; Table 3). Likewise, hs-cTnT concentrations at 0 h and at 3 h were higher in patients who died during follow-up than in survivors (35 (24.5–68.5) vs. 24 (17–46) ng/L, *p* = 0.041 for hs-cTnT at 0 h; 32 (25–62.5) vs. 25 (18–51.3) ng/L, *p* = 0.145, for hs-cTnT at 3 h; Table 3).

Cox proportional hazard models for all-cause death at 365 days revealed that elevated copeptin values carried independent prognostic value (HR 5.37, 95% CI 2.29–12.6, *p* < 0.001) after adjusting for age and sex. This prognostic information was retained after adjustment for hs-cTnT. In contrast, neither hs-cTnT at baseline nor the absolute value at 3 h or the concentration change in hs-cTnT carried independent prognostic information after adjusting for copeptin (Table 4, Figure 3).

### 3.3. Validation Study

The BACC study population consisted of 2719 patients, 1286 patients of whom fulfilled the entry criterion for the present analysis of an available assessment of copeptin. The population considered for the analysis with a diagnosis of NSTE-ACS or acute HF consisted of 288 (201 males, median age 72 years) patients as described in the consort diagram in Appendix A. The baseline characteristics of the entire study group split by diagnostic category are shown in Appendix A and were very similar when compared with the derivation cohort. Overall, 212 cases with NSTE-ACS including type 1 MI and unstable angina with stable elevated hs-cTnT were identified. Copeptin concentrations were higher in patients with acute HF than in the group of NSTE-ACS patients (27.2 (12.8–59.9) vs. 9.6 (4.9–28), *p* < 0.001). Baseline hs-cTnT at 0 h and at 3 h of NSTE-ACS were higher than in patients with acute HF (27 (17–42) vs. 38 (18–121) ng/L at 0 h, *p* = 0.025; 26 (17–50)] vs. 67 (25–201) ng/L at 3 h, *p* < 0.001).

The discrimination of acute HF against type 1 MI was poor for hs-cTnT at baseline (AUC: 0.413, 95% CI: 0.34–0.48, SE: 0.035), at 3 h (AUC: 0.306, 95% CI: 0.24–0.37, SE:0.033; Appendix A), for the concentration difference (AUC: 0.263, 95% CI: 0.21–0.32, SE: 0.028) and for the maximal concentration (AUC: 0.308, 95% CI: 0.24–0.37, SE: 0.033) (Appendix A). For copeptin, the discrimination was moderate (AUC: 0.670, 95% CI: 0.60–0.74, SE: 0.035) and increased non-significantly after combination with hs-cTnT (delta AUC 0.03, *p* = 0.2).

The 1-year mortality rate in the validation cohort was low (33 deaths, 11.5%) and higher in patients with acute HF than in those with type 1 MI (HR: 2.151, 95% CI (1.078, 4.290), *p* = 0.030; 14 vs. 19 deaths; 18.42 vs. 9.98%). Cox regression analysis for all-cause death showed that copeptin predicted 365-day mortality, either as a continuous variable, or at the cutoff that yielded 90% specificity for the discrimination of acute HF. Conversely, hs-cTnT did not predict 365-day mortality, even after log transformation used as a continuous variable or at the 90% specificity cutoff (Appendix A, Appendix A).

## 4. Discussion

Our data show that in situations where natriuretic peptide measurements are not recommended for routine diagnostic assessment, because the chief complaint is suggestive of ACS rather than acute HF, a highly elevated copeptin indicates acute HF rather than uncomplicated NSTEMI. The early differential diagnosis has huge therapeutic implications, and it is pivotal to start the right treatment as soon after presentation as possible. Apart from our findings on copeptin, we have shown that the ability of cardiac troponin to discriminate between myocardial injury and infarction is poor, and optimal thresholds associated with 90% specificity for the diagnosis of acute HF are situated at almost five times the upper limit of normal, i.e., around 70 ng/L. The AUCs vary between 0.460 and 0.441 for the hs-cTnT assessed at 0 h and at 3 h. In contrast, copeptin has demonstrated a better ability to discriminate acute HF, with an AUC of 0.725. The combination of hs-cTnT with copeptin further increases the AUC, albeit in neither a clinically meaningful nor statistically significant way. With regards to prognostic evaluation, we found that a higher risk of all-cause death at 365 days was best predicted by an elevated copeptin but not by hs-cTnT. In an adjusted Cox proportional hazard model, copeptin was associated with a 1% higher risk of death per 1 pmol/L increase at 365 days, which retained its prognostic information after adjustment for hs-cTnT. We validated our observations in an external NSTE-ACS cohort.

Both ACS and acute HF are diagnoses that require immediate attention in the ED. Given that the differentiation between acute myocardial injury and acute MI rests on clinical criteria that suggest a context of myocardial ischemia, an accurate diagnosis may be challenging when patients present with atypical symptoms or dyspnea. In particular, emergency physicians are frequently required to differentiate between type 1 MI, type 2 MI, or an acute myocardial injury due to miscellaneous causes. Although 2D echocardiography and cardiac imaging are strongly recommended for the diagnosis and differential diagnosis of suspected ACS, there is an obvious underuse of cardiac imaging in the ED [24]. Unless acute HF is suspected, natriuretic peptides are not routinely used in suspected ACS and the use of other biomarkers such as copeptin is discouraged in ESC guidelines published in 2020 [8]. The identification of acute HF, however, is pivotal, because the early initiation of treatment with loop diuretics is associated with improved oxygenation [25] and lower in-hospital mortality rates [26].

Copeptin can add important information in this regard. The 39-amino-acid-long peptide is secreted from the pituitary gland stoichiometrically with biologically active vasopressin [27]. The release of copeptin is deemed to be associated with an endogenous stress response. So far, copeptin has been evaluated across a variety of clinical scenarios. Using data from the OPTIMAAL trial, Voors et al. found that copeptin is a strong marker of morbidity and mortality in patients with HF after acute MI [28]. Data from the present study also support data from the BACH trial, in which elevated levels of copeptin predicted increased 90-day mortality, re-admissions and emergency department visits in patients with acute HF [13]. Other studies have not only shown copeptin to predict mortality in patients after an event of worsening HF but have also shown that copeptin can predict re-hospitalization [29]. Likewise, in patients with ACS, previous studies have found that copeptin has independent and additive prognostic value [10,12]. Similar results were found in patients with symptomatic coronary artery disease [30] as well as in those after acute pulmonary embolism [31]. Using data from 2700 patients with symptomatic coronary artery disease who either presented with suspected ACS to the ED or for elective coronary angiography, von Haehling et al. [30] reported that the predictive performance of copeptin was independent of other clinical variables or cardiovascular risk factors, and superior to that of troponin I and other cardiac biomarkers (*p* < 0.0001).

Using data from 1967 patients presenting with chest pain, the CHOPIN trial has shown that the addition of copeptin to the initial work-up, including cTnI assessment, allowed the safe rule-out of acute MI with a negative predictive value >99% in patients with suspected ACS [12]. Compared with cTnI (Chi-square 13.7, c-index 0.828), copeptin (chi-square 29.2, c-index 0.872) had better prognostic value with regards to outcome prediction at 30 days. Recently, the ConTrACS study pooled data from 3890 patients who participated in three observational studies [11]. The study reported the prognostic value of copeptin amongst patients with and without a final ACS diagnosis. The prognostic information was independent of hs-cTnT and the GRACE score. This finding corroborates previous findings on the prognostic role of copeptin in ACS, confirmed PE and acute HF [11,32].

Due to the fast and reversible release kinetics, a dual-marker strategy for the instant rule-out of MI is recommended if copeptin and cardiac troponin are below their respective cutoffs. However, as supported by our data, elevated levels of copeptin carry independent prognostic information, and this finding is irrespective of the underlying condition, which can be extrapolated to acute HF, thus extending our knowledge to an area for which available data have been sparse.

## 5. Limitations

We cannot exclude a sample size error since the numbers of patients with acute HF and fatality rates were low. A power calculation was not performed given the observational nature of the study.

Access to external cohorts for the validation of non-routine biomarkers is challenging. Therefore, a multicenter validation was not in our scope.

In order to validate the preliminary findings of this pilot study, we validated our findings from the test cohort in an external independent NSTE-ACS cohort. Given the low event rate, Cox regression modelling restricts adjustment to only a few possible confounders. Therefore, we cannot exclude that copeptin might lose its independent prognostic value after full adjustment in a considerably larger study population. Copeptin, while recommended for the instant rule-out of MI if a high sensitivity cardiac troponin assay is not available, is still not available on a fully automated central laboratory instrument. This issue has probably hindered a broader market availability.

## Figures and Tables

**Figure 1 cells-12-01062-f001:**
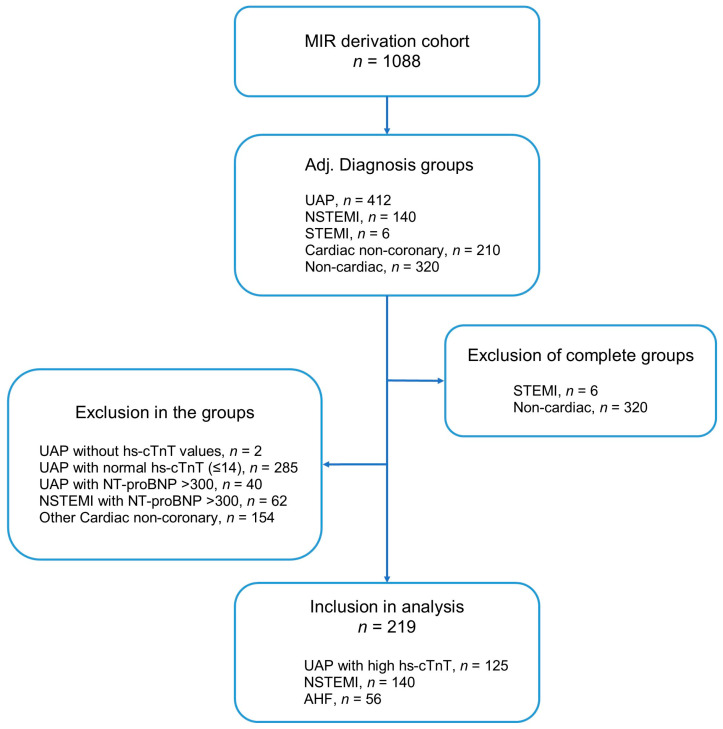
Consort diagram derivation cohort. Abbreviations: AHF: acute heart failure; hs-cTnT: high sensitivity cardiac troponin T; NSTEMI: non-ST-segment elevation myocardial infarction; NT-proBNP: N-terminal pro-B-type natriuretic peptide; STEMI: ST-segment elevation myocardial infarction; UAP: unstable angina pectoris.

**Figure 2 cells-12-01062-f002:**
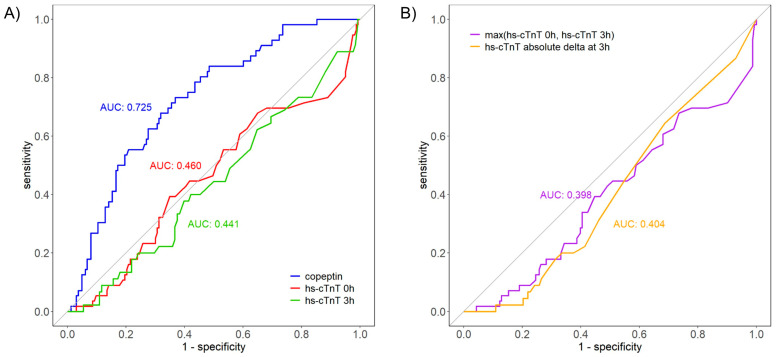
(**A**) ROC comparison of discriminatory ability of copeptin, hs-cTnT at 0 h and hs-cTnT at 3 h in the derivation cohort. (**B**) ROC comparison of discriminatory ability of hs-cTnT as maximal concentration and concentration change between value at presentation and value after 3 h in the derivation cohort.

**Figure 3 cells-12-01062-f003:**
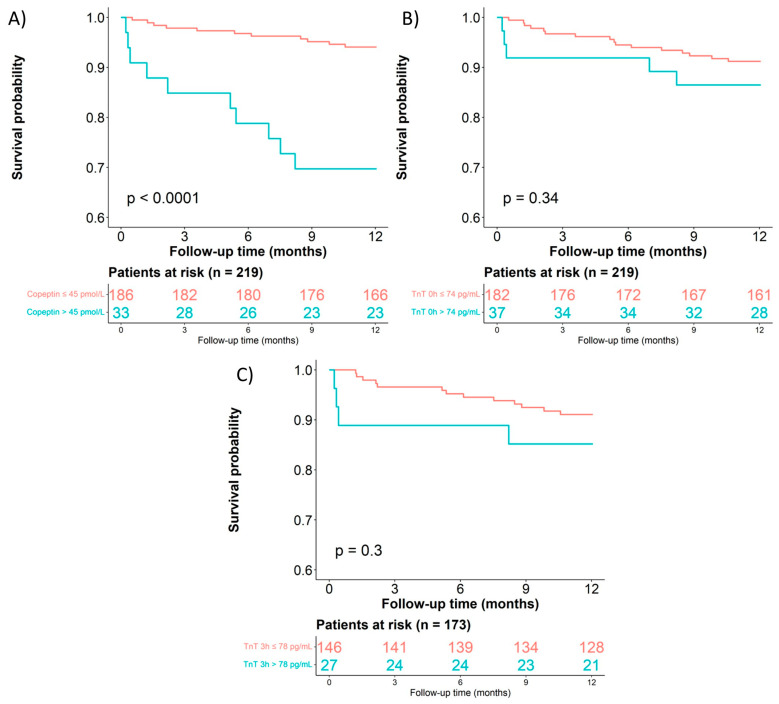
Kaplan–Meier curves. (**A**) Copeptin ≤45 vs. >45 pmol/L by all-cause death at 365 days. (**B**) hs-cTnT at 0 h ≤74 vs. >74 ng/L by all-cause death at 365 days. (**C**) hs-cTnT at 3 h ≤78 vs. >78 ng/L by all-cause death at 365 days.

**Table 1 cells-12-01062-t001:** Baseline characteristics and laboratory values in the derivation cohort.

	All Patients (*n* = 219)	Acute HF (*n* = 56)	NSTE-ACS (*n* = 163)	*p*-Value
Age	70.27 ± 12.41	73.27 ± 12.33	69.24 ± 12.31	0.036
Age > 75 years	96 (44%)	31 (55%)	65 (40%)	0.044
Male (%)	163 (74%)	37 (66%)	126 (77%)	0.097
eGFR (ml/min/1.73 m^2^)	73.2 (51.5–89.9)	53.5 (38.1–76.9)	78.6 (61.7–91.6)	<0.001
eGFR < 60 mL/min/1.73 m^2^	68 (31%)	33 (59%)	35 (26%)	<0.001
eGFR < 30 mL/min/1.73 m^2^	13 (6%)	7 (13%)	6 (4%)	0.016
**CV Risk**				
Hypertension, No. (%)	184 (84%)	50 (90%)	134 (82%)	0.212
Diabetes mellitus, No. (%)	64 (29%)	12 (21%)	52 (32%)	0.117
Hypercholesterolemia, No. (%)	142 (65%)	42 (75%)	100 (61%)	0.065
**Chief complaint**				
Chest Pain, No. (%)	193 (88%)	37 (66%)	156 (96%)	<0.001
Dyspnea, No. (%)	117 (53%)	47 (84%)	70 (43%)	<0.001
**Medical history**				
Previous MI, No. (%)	65 (29.7%)	17 (30.4%)	8 (29.4%)	0.898
Previous PCI/CABG, No. (%)	120 (54.8%)	36 (64.3%)	84 (51.5%)	0.098
History of HF, No. (%)	22 (10.0%)	17 (30.4%)	5 (3.1%)	<0.001
Previous stroke, No. (%)	20 (9.1%)	6 (10.7%)	14 (8.6%)	0.634
GRACE score	115.43 ± 26.16	127.91 ± 26.8	111.09 ± 24.57	<0.001
**Laboratory**				
Copeptin (pmol/L)	9.2 (4.9–17.7)	17.14 (8.8–54.5)	7.7 (4–14.5)	<0.001
Hs-cTnT 0 h (ng/L)	26 (17–47)	25.5 (15–40.8)	26 (18–47)	0.373
Hs-cTnT 3 h (ng/L)	27 (18–50.5)	24 (16–37)	27.5 (18–53.5)	0.236
Hs-cTnT delta	3 (1–8)	2 (1–4)	3 (1–11)	0.054
Maximum hs-cTnT	28 (19–64)	25.5 (16–40.8)	29 (19–73)	0.022
**NT-proBNP (admission)** **pg/mL**	283 (134–3114) (*n* = 111)	3287 (2155–6250) (*n* = 53)	135 (52–204) (*n* = 58)	<0.001

Abbreviations: m: male; eGFR: estimated glomerular filtration rate; CV: cardiovascular; MI: myocardial infarction; PCI: percutaneous coronary intervention; CABG: coronary artery bypass grafting; HF: heart failure; hs-cTnT: high sensitivity cardiac troponin T.

**Table 2 cells-12-01062-t002:** Median serum concentration with respect to 99th percentile cut-off value.

	Group	Median within Group	Median Below or Equal Cut-Off	Median Above Cut-Off
**Copeptin ** **cut-off = 10 pmol/L**	All	9.2 (4.9–17.7) *n* = 219	5.2 (3.4–7.5) *n* = 119 (54.3%)	21.3 (14.04–56.03) *n* = 100 (45.7%)
	Acute HF	17.14 (8.8–54.5) *n* = 56	5.8 (4.5–8.6) *n* = 16 (28.6%)	28.5 (16.2–57.3) *n* = 40 (71.4%)
	NSTE-ACS	7.7 (4–14.5) *n* = 163	5.0 (3.3–6.95) *n* = 103 (63.2%)	16.9 (13.1–48.2) *n* = 60 (36.8%)
**Hs-cTnT 0 h** **cut-off = 14 ng/L**	All	26 (17–47) *n* = 219	12 (10–13) *n* = 19 (8.68%)	27 (18.3–50.3) *n* = 200 (91.32%)
	Acute HF	25.5 (15–40.8) *n* = 56	13 (10–13) *n* = 11 (19.6%)	31 (21.5–49) *n* = 45 (80.4%)
	NSTE-ACS	26 (18–47) *n* = 163	11.5 (8.5–13) *n* = 8 (4.9%)	27 (18–52) *n* = 155 (95.1%)
**Hs-cTnT 3 h** **cut-off = 14 ng/L**	All	27 (18–50.5) *n* = 173	13 (12–14) *n* = 15 (8.7%)	28 (19.8–55) *n* = 158 (91.3%)
	Acute HF	24 (16–37) *n* = 45	12 (11.5–12.5) *n* = 5 (11.1%)	26.5 (19.3–42.5) *n* = 40 (88.9%)
	NSTE-ACS	27.5 (18–53.5) *n* = 128	14 (12.75–14) *n* = 10 (7.8%)	28 (20.5–56) *n* = 118 (92.2%)

**Table 3 cells-12-01062-t003:** Baseline characteristics and laboratory values by survival status.

	All Patients (*n* = 219)	Death (*n* = 21)	Alive (*n* = 198)	*p*-Value
Age	70.27 ± 12.41	79.05 ± 10.23	69.34 ± 12.28	<0.001
Age > 75 years	96 (44%)	17 (81%)	79 (40%)	<0.001
m/f	163/56 (74%/26%)	15/6 (71%/29%)	148/50 (75%/25%)	0.740
aHF/NSTE-ACS	56/163 (25.6%/74.4%)	15/6 (71.4%/28.6%)	41/157 (20.7%/79.3%)	<0.001
GRACE score	115.43 ± 26.16	138.62 ± 0	112.94 ± 138.62	<0.001
eGFR	73.2 (51.5–89.9)	54 (25.9–66.8)	76.5 (56–91.3)	<0.001
eGFR < 60 mL/min	68 (31%)	14 (67%)	54 (27%)	<0.001
eGFR < 30 mL/min	13 (6%)	6 (29%)	7 (4%)	<0.001
**Laboratory**				
Copeptin	9.2 (4.9–17.7)	31.6 (12.3–63.8)	8.5 (4.4–16.1)	<0.001
Hs-cTnT 0 h	26 (17–47)	35 (24.5–68.5)	24 (17–46)	0.041
Hs-cTnT 3 h	27 (18–50.5)	32 (25–62.5)	25 (18–51.3)	0.145
Hs-cTnT delta	3 (1–8)	3 (1–11.5)	3 (1–8)	0.894
Maximum hs-cTnT	28 (19–64)	37 (27–68.5)	28 (18–64.5)	0.185

**Table 4 cells-12-01062-t004:** Cox regression models for prediction of all-cause death at 365 days.

	Univariable Model		Multivariable Model 1 *		Multivariable Model 2 **	
	HR (95% CI)	*p*-Value	HR (95%CI)	*p*-Value	HR (95% CI)	*p*-Value
**Copeptin (log)**	6.50 (2.82–15.01)	<0.001	5.4 (2.29–12.6)	<0.001	4.7 (1.9–11.45)	0.001
**Hs-cTnT 0 h (log)**	1.78 (0.78–4.07)	0.174			1.5 (0.59–3.92)	0.384
**Hs-cTnT 3 h (log)**	1.39 (0.57–3.38)	0.465				
**Age**	1.10 (1.04–1.16)	0.001	1.08 (1.03–1.1)	0.003	1.1 (1.03–1.15)	0.005
**Male**	0.85 (0.33–2.19)	0.736	0.88 (0.34–2.3)	0.802	0.84 (0.32–2.2)	0.729

* Model 1 was adjusted for copeptin, age and sex. ** Model 2 was adjusted for copeptin, hs-cTnT, age and sex.

## Data Availability

The main study results have been published previously and are cited as reference #16 for the MIR study and #20 and #21 for the BACC study.

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
