# Peer review of "Role of Copeptin and hs-cTnT to Discriminate AHF from Uncomplicated NSTE-ACS with Baseline Elevated hs-cTnT—A Derivation and External Validation Study"

_cells, 2023, doi:10.3390/cells12071062_

Round 1

Reviewer 1 Report

 The article discusses the prognostic properties of copeptin in acute settings and patients with ACS. The article is interesting and suitable for publication. Adding a section listing abbreviations used in the article would be better. Figure 1 footnote should also state important abbreviations used and some description. Brief definitions or explanations of terms used in the article, such as Type 1 MI and Type 2 MI can be added where necessary.

Author Response

Response to Reviewer 1:

The article discusses the prognostic properties of copeptin in acute settings and patients with ACS. The article is interesting and suitable for publication.

  1. Adding a section listing abbreviations used in the article would be better.
  2. Figure 1 footnote should also state important abbreviations used and some description.

Abbreviations were added at Figure 1 and a list of abbreviation is not in the recommendations list for manuscript preparation. Therefore we added the abbreviations where appropiate and at first appearance.

Abbreviations: AHF: acute heart failure; NSTEMI: non-ST-segment elevation myocardial infarction; NT-proBNP:  STEMI: ST-segment elevation myocardial infarction; UAP: unstable angina pectoris.

  1. Brief definitions or explanations of terms used in the article, such as Type 1 MI and Type 2 MI can be added where necessary.

    We thank the reviewer for this comment and we added the explanations for Type 1 MI and Type 2 MI in the Methods section.

    *Type 2 MI was diagnosed in the presence of a rise and/or fall of cTn values with at least one value above the 99th percentile URL, and evidence of inbalance between myocardial oxygen supply and demand unrelated to coronary thrombosis, requiring at least one clinical variable that suggests a context of myocardial eschemia. In contrast, Type 1 MI represents the typical spontaneous MI characterized by plaque rupture/-erosion with occlusive or non-occlusive thrombosis
    .

Reviewer 2 Report

Authors of ms cells-2229682 present the results of retrospective analyses of a derivation and a validation cohort of patients admitted to the University Hospital in Heidelberg in the years 2014 tru 2017. The main result s that highly elevated copeptin concentration, but not hscTnT contribute significantly to the differential diagnosis between ACS and acute HF.

The ms is well written and arguments well taken, given the well known expertise of the authors in the field.

3 comments:

(1) at line 62 the authors write “…suspected ACS, a setting where natriuretic peptides are not recommended routinely.” This is even more the case for copeptin, therefore, the lack of data on NT-proBNP in the present study is a substantial limitation.

(2) the AUC of copeptin is not sufficient to convince clinicians to use it: indeed the assay has not yet been introduced into clinical chemistry lab as automated assay, even after several favorable studies.

(3) the study would have been more convincing if the validation had been done prospectively and in more than one clinical center.

Author Response

Response to Reviewer 2:

Authors of ms cells-2229682 present the results of retrospective analyses of a derivation and a validation cohort of patients admitted to the University Hospital in Heidelberg in the years 2014 tru 2017. The main result s that highly elevated copeptin concentration, but not hscTnT contribute significantly to the differential diagnosis between ACS and acute HF.

The ms is well written and arguments well taken, given the well known expertise of the authors in the field.

3 comments:

(1) at line 62 the authors write “…suspected ACS, a setting where natriuretic peptides are not recommended routinely.” This is even more the case for copeptin, therefore, the lack of data on NT-proBNP in the present study is a substantial limitation.

We thank the reviewer for this comment. We already addressed this shortcoming in the limitation section. In our opinion, copeptin could play a more dominant role in the diagnosis of ACS as suggested by current guidelines that support the measurement of copeptin if hs-cTn is not available. In addition, there is incremental evidence, that copeptin carries added and independent prognostic information in ACS and non ACS patients (Waldsperger et al., Ref 11 in manuscript). Our data are consistent with these previous findings and should stimulate broader use of dual marker strategy (DMS) in ACS, particularly when the leading symptoms are not typical chest pain.

  1. Waldsperger H, Biener M, Stoyanov KM, Vafaie M, Katus HA, Giannitsis E, Mueller-Hennessen M. Prognostic Value of Elevated Copeptin and High-Sensitivity Cardiac Troponin T in Patients with and without Acute Coronary Syndrome: The ConTrACS Study. J Clin Med 2020, 9(11), 3627. doi: 10.3390/jcm9113627. PMID: 33187192; PMCID: PMC7696893.

(2) the AUC of copeptin is not sufficient to convince clinicians to use it: indeed the assay has not yet been introduced into clinical chemistry lab as automated assay, even after several favorable studies.

The reviewer is correct but still DMS with copeptin represents an alternative to the recommended 0/1hr algorithm. If DMS is used, the ability to discriminate acute heart failure without NSTEMI from NSTEMI without heart failure is complimentary and supports the broader use of copeptin. In support of the versatile usefulness of copeptin, Neumann et al. (Reference below) found that copeptin was one of four biomarkers that were able to discriminate between Type 1 and Type 2 MI.

A Biomarker Model to Distinguish Types of Myocardial Infarction and Injury. Neumann JT, Weimann J, Sörensen NA, Hartikainen TS, Haller PM, Lehmacher J, Brocks C, Tenhaeff S, Karakas M, Renné T, Blankenberg S, Zeller T, Westermann D. J Am Coll Cardiol. 2021 Aug 24;78(8):781-790. doi: 10.1016/j.jacc.2021.06.027. PMID: 34412811

(3) the study would have been more convincing if the validation had been done prospectively and in more than one clinical center.

We agree with the reviewer that prospective validation in a multicenter study, ideally covering different continents and regions and reflecting real world evidence would have been the optimal validation form. However, access to external validation cohorts is difficult, particularly when non-routine biomarkers such as copeptin have to be validated. We expanded this shortcoming in the limitations.

*Access to external cohorts for validation of non-routine biomarkers is challenging. Therefore, a multicenter validation was not in our scope.

Reviewer 3 Report

The submitted manuscript deal with a very important topic. Since introduction of hs-TNT and the Universal definition of MI physician are very confused. From this time point MI is no more exclusively a coronary disease, but the reflex to send every MI to the cath lab based on the Universal definition is still predominant.

To differ between myocardial causes and coronary causes of MI is therefore key. The present data are a further important stone in this building.  Well defined patient population are included and an adequate statistic is used. Limitations are expressed. 

line 73 Ref 8: Please give more specific citation

line 108 as the score did never enter clinical practice please describe. Which cut-point do you use (6?)

figure 1 : please spell in full UAP

line 185. ..heterogenous… I would kindly ask for rephrasing.

NT-proBNP should be present in the patient populations. Please describe the populations also at least in the demographics by this key variable.

I did not found any information, whether every patients received coronary angiography to confirm or exclude CAD. Please comment.

In Ref 29 van Haehling pointed out that in patients with increased copeptin levels there is an increased risk of following MI. Thus, there are two questions.:

1.     Can this be confirmed by the present data?

2.     If this holds true, please discuss a procedure how to deal with patients with elevated Copeptin and increased Troponin?  Would you presume that decompensation is the leading cause and angiography is not necessary or would you just change the time frame of Coronary angiography. I think this are exciting questions in such an important topic, to be discussed.

Please make more clear that this is a population with a high pretest probability for ACS.

You excluded patients with NT-proBNP >300pg/ml. Would you handle those patients as primarily heart failure patients? Please argue more precise, why you excluded those patients. Is my understanding correct that your approach is as follows:  If NT-proBNP > 300 its acute heart failure, if NT-proBNP<300 but Copeptin is elevated its still acute heart failure if Copeptin and Nt-proBNP is low and only hs-TNT is high its ACS? I am aware that the present data are not ready for clinical use, but the reader is interested where the journey might take us.

Author Response

Response to Reviewer 3:

The submitted manuscript deal with a very important topic. Since introduction of hs-TNT and the Universal definition of MI physician are very confused. From this time point MI is no more exclusively a coronary disease, but the reflex to send every MI to the cath lab based on the Universal definition is still predominant.

To differ between myocardial causes and coronary causes of MI is therefore key. The present data are a further important stone in this building.  Well defined patient population are included and an adequate statistic is used. Limitations are expressed. 

line 73 Ref 8: Please give more specific citation

We thank the reviewer for the suggestion. We added the reference Wiviott et al.

Pathophysiology, prognostic significance and clinical utility of B-type natriuretic peptide in acute coronary syndromes. Wiviott SD, de Lemos JA, Morrow DA. Clin Chim Acta. 2004 Aug 16;346(2):119-28. doi: 10.1016/j.cccn.2004.04.004. PMID: 15256312

line 108 as the score did never enter clinical practice please describe. Which cut-point do you use (6?)

In clinical practice, the routine use of GRACE score is very heterogeneous. In our CPU the measurement of the GRACE score was encouraged but not obligatory. For research purpose, the GRACE score was retrospectively calculated for all patients. A GRACE score <108 points indicated low risk and a GRACE score >= 140 points indicated high risk. We added following sentence in the Methods section.

 *The score was used as a continuous variable to characterize the overall risk of NSTEMI or AHF cohort, and was not used to categorize individual patients into low, intermediate or high risk.

figure 1 : please spell in full UAP

We thank the reviewer and explain abbreviations where they first appear.

line 185. ..heterogenous… I would kindly ask for rephrasing.

We have rephrase as
*was heterogenous differed widely according to whether troponin or copeptin were used and by time point of blood draw.

NT-proBNP should be present in the patient populations. Please describe the populations also at least in the demographics by this key variable.

I did not found any information, whether every patients received coronary angiography to confirm or exclude CAD. Please comment.

We added information on rates of coronary angiography by study center and ACS subgroup in the Results, line 166. In addition, we added median NT-proBNP concentrations in Table 1.

In Ref 29 van Haehling pointed out that in patients with increased copeptin levels there is an increased risk of following MI. Thus, there are two questions.:

  1. Can this be confirmed by the present data?
    This is an interesting question that we cannot answer because our follow up could not guarantee the complete assessment of MI’s and MI subtypes. The latter would also require full access to patient information from external hospitals. Therefore we decided to report only all cause death as the most reliable endpoint. Follow up was almost complete (>99%) for vital status.
  2. If this holds true, please discuss a procedure how to deal with patients with elevated Copeptin and increased Troponin?  Would you presume that decompensation is the leading cause and angiography is not necessary or would you just change the time frame of Coronary angiography. I think this are exciting questions in such an important topic, to be discussed.
    This question is very important but cannot be answered by our study findings. Given the additive and independent prognostic information of copeptin and troponin more intense monitoring for clinical deterioration should be employed. If elevated concentrations of copeptin and troponin would prompt earlier angiography is illusive, yet.

Please make more clear that this is a population with a high pretest probability for ACS.

Along with troponin, suspicion of ACS was based on a combination of clinical symptoms, presence of atherosclerotic risk factors, ECG, previous history of CAD/previous MI/previous PCI or CABG.  Patients were admitted to Chest Pain Units in Heidelberg or Hamburg. We added text to the Methods section to clarify on the pretest probability for ACS.

You excluded patients with NT-proBNP >300pg/ml. Would you handle those patients as primarily heart failure patients? Please argue more precise, why you excluded those patients. Is my understanding correct that your approach is as follows:  If NT-proBNP > 300 its acute heart failure, if NT-proBNP<300 but Copeptin is elevated its still acute heart failure if Copeptin and Nt-proBNP is low and only hs-TNT is high its ACS? I am aware that the present data are not ready for clinical use, but the reader is interested where the journey might take us.

In the NSTE-ACS cohort we used NT-proBNP at the sex independent rule-out cut-off. Patients with values <300 pg/ml are unlikely to have an acute heart failure. Contrary we excluded patients with values >300 pg/ml.

In the acute HF cohort, the diagnosis was based on symptoms with NT-proBNP values exceeding the age-dependent cut-off, i.e. >900, >1200, >1800 pg/ml for patients older than 50, 50-75, and > 75 years, respectively.

We clarified this process in more details in the Methods section.